# Robust Prototype Completion for Incomplete Multi-view Clustering

Honglin Yuan
Southwest University of Science and
Technology
Mianyang, China
hl_yuan0822@163.com

Shiyun Lai
Southwest University of Science and
Technology
Mianyang, China
lsy045890@163.com

Xingfeng Li
Nanjing University of Science and
Technology
Nanjing, China
lixingfeng@njust.edu.cn

Jian Dai
Southwest Automation Research
Institute
Mianyang, China
daijian1000@163.com

Yuan Sun*
Sichuan University
Chengdu, China
sunyuan_work@163.com

Zhenwen Ren*
Southwest University of Science and
Technology
Mianyang, China
rzw@njust.edu.cn

## Abstract

In practical data collection processes, certain views may become partially unavailable due to sensor failures or equipment issues, leading to the problem of incomplete multi-view clustering (IMVC). While some IMVC methods employing prototype completion achieve satisfactory performance, almost all of them implicitly assume correct alignment of prototypes across all views. However, during prototype generation, different networks could generate different cluster centers, thereby leading to the produced prototypes from different views may be misaligned, *i.e.*, prototype noisy correspondence. To address this issue, we propose Robust Prototype Completion for Incomplete Multi-view Clustering (RPCIC), which mitigates the impact of noisy correspondence in prototypes. Specifically, RPCIC initially utilizes cross-view contrastive learning module to obtain consistent feature representations across different views. Subsequently, we devise robust contrastive loss for the produced prototypes, aiming to alleviate the influence of noisy correspondence within them. Finally, we employ prototype fusion-based strategy to complete the missing data. Comprehensive experiments demonstrate that RPCIC outperforms 11 state-of-the-art methods in terms of both performance and robustness. The code is available at https://github.com/hl-yuan/RPCIC.

## CCS Concepts

• **Theory of computation → Unsupervised learning and clustering**.

*Corresponding author

## Keywords

Incomplete multi-view clustering, prototype completion, prototype noisy correspondence.

**ACM Reference Format:**
Honglin Yuan, Shiyun Lai, Xingfeng Li, Jian Dai, Yuan Sun, and Zhenwen Ren. 2024. Robust Prototype Completion for Incomplete Multi-view Clustering. In *Proceedings of the 32nd ACM International Conference on Multimedia (MM '24), October 28-November 1, 2024, Melbourne, VIC, Australia.* ACM, New York, NY, USA, 10 pages. https://doi.org/10.1145/3664647.3681397

## 1 Introduction

Multi-view data widely exists in practical applications, where is usually collected from various sources with heterogeneous attributes [35]. As a fundamental unsupervised learning technique, multi-view clustering (MVC) aims to partition multi-view data into the corresponding categories by utilizing the consistent and complementary information from different views. In recent years, numerous MVC methods have been proposed to obtain the promising clustering performance [1, 22, 42, 43, 46], which become a hot topic in the machine learning community. Existing MVC methods rely heavily on the data completeness assumption. However, such an assumption can easily be violated in real-world scenes. In other words, some instances could contain only partial views in multi-view data due to unstable or damaged sensors, which inevitably results in the incomplete multi-view clustering (IMVC) problem. Code is a

In recent years, many IMVC methods [16, 23, 27, 39] have been proposed, which adopt the observed samples to identify cross-view neighbors, thereby imputing the missing data. They usually project all views into a common space, thereby seeking neighbors corresponding to the missing samples. Nevertheless, to learn the consistency of different views, these methods often neglect the view-specific information, thereby sacrificing the view versatility. To this end, some methods adopt generator [12, 14, 45] and predictor [25, 26] for missing data imputation, which can capture the diversity of all views. Unfortunately, they tend to learn the specific representations from all views, which will result in losing the consensus of the instances. To preserve the consistency and specificity of all views, some prototype-based methods [7, 9, 11] construct the sample-prototype relationships between the observation samples and the prototypes to recover the missing multi-view data. Most

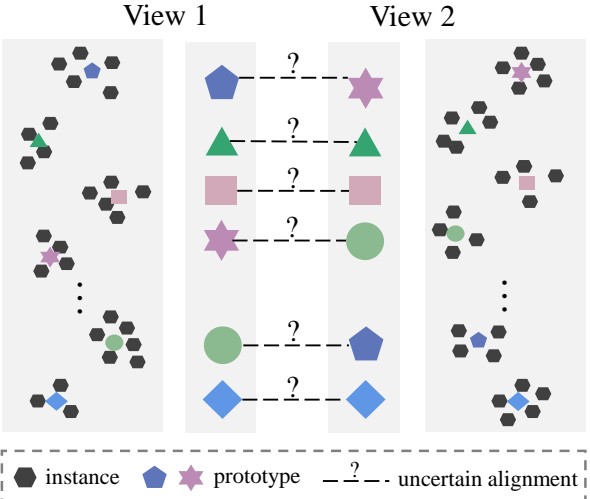

**Figure 1: Examples of prototype noisy correspondence. Due to prototypes being generated independently for each view, and variations in prototype quality across different views, the learned prototypes from different views could be not perfectly aligned. In this scenario, some misaligned prototypes may mistakenly be considered aligned, thus resulting in the problem of prototype noisy correspondence.**

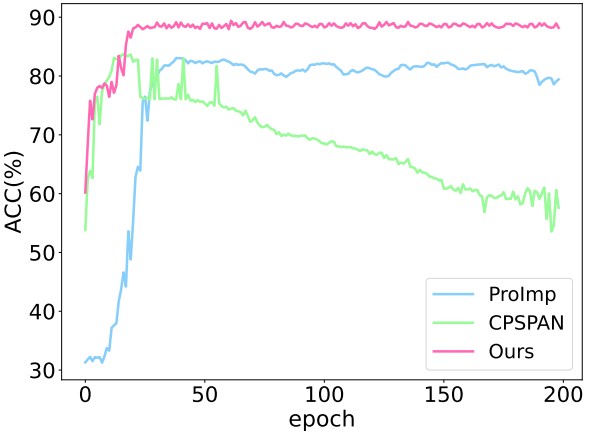

**Figure 2: The ACC performance of different methods on the HandWritten with the missing rate 0.9 as the number of epochs increases.**

of them implicitly assume that the learned prototypes from different views are perfectly aligned. However, different networks could produce different clustering centers, resulting in prototype misalignment across different views (see in Fig.1), *i.e.*, prototype noisy correspondence [6, 10, 20, 21]. Clearly, misalignment prototypes will mislead the model, thereby significantly reducing clustering performance. Specifically, as shown in Fig.2, under the high missing rate (*i.e.*, 0.9), CPSPAN [7] overfits the false positives. Although ProImp [9] has more robustness, it suffers from underfitting.

To overcome the aforementioned issues, we propose a novel Robust Prototype Completion for Incomplete Multi-view Clustering

(RPCIC), as shown in Fig.3. Specifically, we first construct the positive and negative pairs and adopt cross-view contrastive learning (CCL) to maximize multi-view consistency, thereby narrowing the multi-view heterogeneous gap. To overcome the adverse impact caused by prototype noisy correspondence in multi-view learning, we propose a robust prototype discriminative learning (RPD) loss to robustly learn from prototype noisy correspondence, thereby mitigating the overfitting and underfitting problems. Finally, to recover missing multi-view data, we propose a prototype fusion-based completion strategy. In general, the main contributions of this paper are as follows:

- We propose a novel incomplete multi-view learning framework named RPCIC. Unlike existing prototype-based methods, our RPCIC could be the first work to reveal and study prototype noisy correspondence in prototypes for multi-view learning.
- To tackle the overfitting problem caused by prototype noisy correspondence, we propose a robust prototype discriminative learning loss to resist the noise interference from false positives, thus embracing robustness.
- We theoretically and experimentally demonstrate the robustness of our RPCIC. Numerous experiments show that RPCIC remarkably outperforms 11 state-of-the-art comparison methods on four datasets with different missing rates.

## 2 Related Work

### 2.1 Incomplete Multi-view Clustering

Over the past decade, a series of IMVC methods have been proposed [31–33, 36, 44]. Classical IMVC methods can be classified into three categories: matrix factorization-based methods [17, 28, 48], kernel-based methods [18, 19], and graph learning-based methods [11, 29, 30]. Thanks to strong non-linear representation ability, some deep IMVC methods have been developed in recent years, which has gradually become a mainstream research direction. According to the way of missing data imputation, these DIMVC methods can be classified into four categories: (1) Predictor-based methods [12, 14], which predicts missing views from observed views and recover data by minimizing conditional entropy. (2) GAN-based methods [25, 26], which adopt the generative adversarial network (GAN) to recover missing multi-view data. (3) Neighborhood-based methods [23, 39], which directly impute missing data by finding nearest neighbors across different views. (4) Prototype-based methods [7, 9], which learn prototypes from the missing view and explore the sample-prototype relationships in the observed view to recover data. Although prototype-based methods obtain promising clustering performance, the learned prototypes of each cluster from different views inevitably have some deviations, thereby resulting in prototype noisy correspondence. Therefore, there is an urgent need for an IMVC method to mitigate the negative effects of noise.

### 2.2 Contrastive Learning

Due to the excellent performance of contrastive learning in the unsupervised field, MVC tasks have also widely adopted this method to obtain information about the data between different views. The core concept of contrastive learning is to maximize the similarity between positive sample pairs and minimize the similarity between

negative sample pairs in the potential space. For instance, COM-PLETER [15] proposes a cross-view contrastive learning paradigm, which aims to maximize the mutual information between different views to obtain information-rich consistent representations. To further enhance the mining capability of consistent information, MCMVC [3] and Dealmvc [41] employ a dual-contrast strategy, utilizing instance-level and class-level contrastive learning to obtain consistent representations of samples. However, the performance of traditional contrastive learning methods can be affected by false negative pairs. To address this issue, SURE [39] proposes a novel contrastive loss and employs a two-stage optimization scheme. Different from these methods, we reveal the problem of noisy correspondence in prototypes and propose a robust prototype completion framework, thereby enhancing the robustness against false negative pairs.

## 3 Method

### 3.1 Notations

Let $X^v \in \mathbb{R}^{N \times G_v}$ as the incomplete multi-view dataset with $V$ views, where $G_v$ represents the feature dimensionality in $v$-th view and $N$ is the total number of instances. $\tilde{X}^v \in \mathbb{R}^{N_v \times G_v}$ denotes the complete dataset, where $N_v$ represents the number of all instances from each view. $H^v \in \mathbb{R}^{N_v \times d}$ represents the view-specific representations of the $v$-th view, where $d$ denotes the embedding dimensionality. Similar to [7], for any two views, we define the pair-observed representations in the complete data as $Z^v$ and the unpaired representation as $U^v$. IMVC aims to divide incomplete multi-view data with the same semantics into $C$ clusters.

### 3.2 The Objective Function

Our proposed RPCIC mainly contains two stages, *i.e.*, warm-up and fine-tune. In the first stage, we adopt within-view reconstruction loss to pre-train the model, thereby preserving as much as possible view information independently. In the second stage, we first utilize cross-view contrastive learning to explore consistent representations across different views. To complete the missing multi-view data through learning a common representation, we further generate view-specific prototypes. However, this inevitably causes the noisy correspondence problem in the generated prototype set, which could cause the model to overfit hard samples. To this end, we propose a robust discriminative learning loss to reduce the focus on noisy data. Finally, we propose a prototype fusion-based imputation strategy to recover missing data. In general, the objective loss function of RPCIC is formulated as follows:

$$\mathcal{L} = \mathcal{L}_{rec} + \alpha \mathcal{L}_{ccl} + \beta \mathcal{L}_{rpd}, \tag{1}$$

where $\alpha$ and $\beta$ are the trade-off parameters. $\mathcal{L}_{rec}$, $\mathcal{L}_{ccl}$ and $\mathcal{L}_{rpd}$ are reconstruction loss, cross-view contrastive loss, and robust prototype discriminative loss.

### 3.3 Within-view Reconstruction

To learn the clustering-friendly features for each view, we adopt the widely used deep view-specific autoencoder to obtain high-level feature representations. Specifically, we adopt the encoder $E_v$ to encode the complete data $\tilde{X}^v \in \mathbb{R}^{N_v \times G_v}$ into view-specific representations $Z^v \in \mathbb{R}^{N_v \times d}$, where $d$ is the dimensionality of the

representations on all views. Then, we use the decoder $D_v$ to decode $Z^v$ into $\hat{X}^v \in \mathbb{R}^{N_v \times G_v}$. Overall, the within-view reconstruction loss can be defined as follows:

$$\mathcal{L}_{rec} = \sum_{v=1}^{v} \mathcal{L}_{rec}^v = \sum_{v=1}^{V} \left\| \tilde{X}^v - \hat{X}^v \right\|_F^2, \tag{2}$$

where $H^v = E_v(\tilde{X}^v; \theta^v)$ and $\hat{X}^v = D_v(H^v; \phi^v)$. $\theta^v$ and $\phi^v$ represent the network parameters of the encoder and decoder for the $v$-th view, respectively.

### 3.4 Cross-view Contrastive Learning

To alleviate the cross-view heterogeneity gap in multi-view data, we adopt cross-view contrastive learning (CCL) to maximize the consistency between positive pairs while minimizing that between negative pairs. Specifically, we use the cosine distance to measure the similarities between cross-view representations as follows:

$$S(z_i^v, z_j^u) = \frac{z_i^v (z_j^u)^\top}{\|z_i^v\| \|z_j^u\|}, \tag{3}$$

where $z_i^v$ and $z_j^u$ are the pair-observed representations. To learn discriminative representations, we adopt the classical contrastive loss InfoNCE [24], *i.e.*,

$$\mathcal{L}_{ccl}^{vu} = -\frac{1}{N_v} \sum_{i=1}^{N_v} \log \frac{\exp^{\left(S(z_i^v, z_i^u)/\tau\right)}}{\sum_{j=1}^{N_v} \exp^{\left(S(z_i^v, z_j^u)/\tau\right)}}, \tag{4}$$

where $\tau$ denotes a temperature parameter and is set to 1. The overall loss function across all views can be formulated as:

$$\mathcal{L}_{ccl} = \sum_{v=1}^{V} \sum_{u \neq v}^{V} \mathcal{L}_{ccl}^{vu}. \tag{5}$$

### 3.5 Robust Prototype Discriminative Learning

Our RPCIC mainly utilizes prototypes to capture diversity and consistency from different views, thereby recovering the missing data. Therefore, we further generate a set of prototypes for each view. Specifically, we first denote the prototype rate $m$ as the ratio of the number of prototypes to the minimum number of observed instances. Then adopt $k$-means clustering to get a certain percentage (*i.e.*, $m$) of cluster centers. Hence, we can obtain $K$ prototypes. And we treat the view-specific representations with the smallest distance from each cluster center as prototypes $P$ of each view. Due to insufficient data, different networks could produce different clustering centers, thereby inevitably leading to prototype misalignment across different views, *i.e.*, prototype noisy correspondence.

To overcome this issue, we propose a robust prototype discriminative learning loss to rectify the noisy correspondence in the cross-view prototypes. Specifically, we first calculate cosine similarity to measure the similarity of prototype structures across different views as follows:

$$S(p_i^v, p_j^u) = \frac{p_i^v (p_j^u)^T}{\|p_i^v\| \|p_j^u\|}. \tag{6}$$

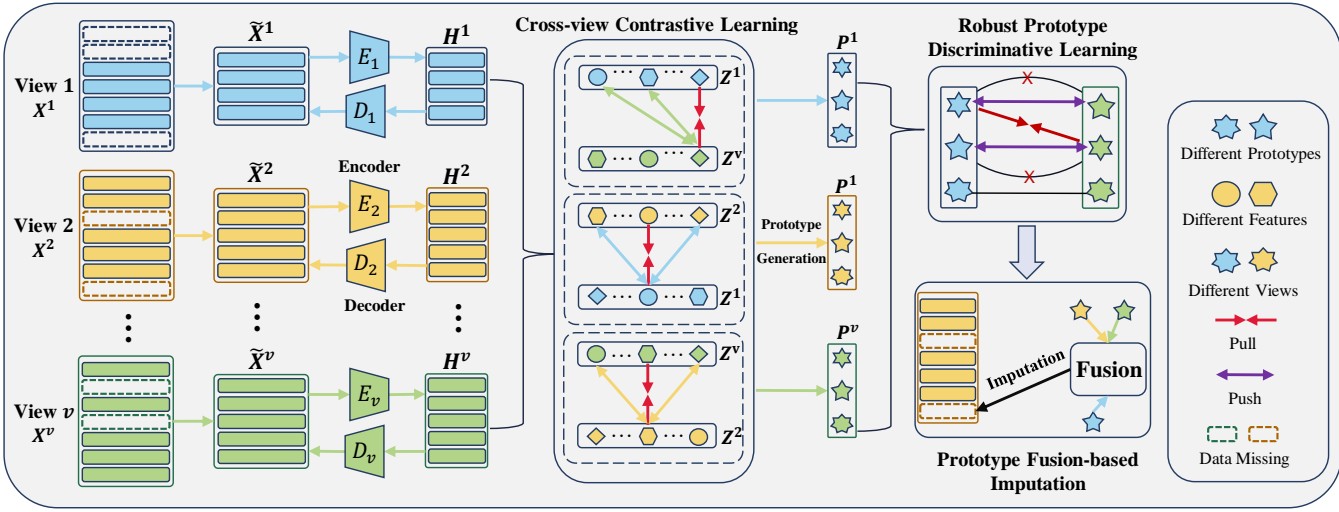

**Figure 3: The framework of the proposed RPCIC. We first adopt within-view reconstruction to learn view-specific representations $H$ from the complete data $\tilde{X}$. Then, to explore the consistency of the pair-observed representations $Z$, we adopt cross-view contrastive learning to reduce the heterogeneous gap. To alleviate the misleading of noisy correspondence in the prototypes $P$, we propose a robust prototype discriminative learning loss. Finally, to recover the missing data, we design a prototype fusion-based imputation strategy to enhance the clustering performance.**

Then, we calculate the probability that the cross-view samples belong to the same instance as follows:

$$Q_i^{vu} = \frac{e^{(S(p_i^v, p_i^u)/\tau)}}{\sum_{j=1}^K e^{(S(p_i^v, p_j^u)/\tau)}}, \qquad (7)$$

where $\tau$ is set to 1. Some studies [47] have shown the loss InfoNCE [24] pays more attention to hard sample pairs, which are false positives, thereby leading to the overfitting problem. To improve robustness, the loss MAE [5] treats all samples equally. However, it inevitably lacks the ability to focus on information pairs, thus leading to the underfitting issue. To overcome these problems, we propose a robust contrastive loss to prevent overemphasizing noisy prototype pairs. Mathematically,

$$\mathcal{L}_{rpd}^{vu} = \frac{1}{K} \sum_{i=1}^K (1 - Q_i^{vu})^r (-\log Q_i^{vu})^{(1-r)}, \qquad (8)$$

where $r \in [0, 1]$ is a hyper-parameter. Finally, the overall loss function $\mathcal{L}_{rpd}$ could be written as:

$$\mathcal{L}_{rpd} = \sum_{v=1}^V \sum_{u \neq v}^V \mathcal{L}_{rpd}^{vu}. \qquad (9)$$

### 3.6 Prototype Fusion-based Imputation

Once the view-specific representations are learned, we adopt the category-level alignment scheme [40] to realign the prototypes from different views. Specifically, we compute the cross-view smallest euclidean distance matrix of the learned prototypes, thereby achieving prototype realignment.

Prototypes possess shared characteristics of heterogeneous multi-view data, which could capture the cluster- and view-specific information to recover missing data. In addition, since the number

of missing data from different views is different, simply using the prototypes of a view to recover the missing data could introduce bias into the entire data structure. To this end, we propose the prototype fusion-based imputation strategy to alleviate the biased impact of the prototype imputation from a single view. As shown in Fig.4, considering the bi-view data, we first calculate the sample-prototype similarities for each view. Then, for a missing sample in a view, we use cross-view structure matching to locate its corresponding neighbor sample in another view. According to the sample-prototype similarities, we find the corresponding prototype and obtain the matching prototype from another view. Finally, the fusion prototypes can be obtained by the following formula:

$$F_i^v = \frac{1}{V} \sum_{v=1}^V P_i^v, \qquad (10)$$

where $F_i^v$ denotes the fusion prototypes. Afterward, we use the fusion prototypes to recover this missing sample. After the whole missing data is recovered, we obtain a common representation by concatenating whole data from all views and adopt $k$-means for it to achieve IMVC.

### 3.7 Robustness Analysis

The goal of our loss is to inherit the advantages of both InfoNCE (*i.e.*, $r = 1$) and MAE (*i.e.*, $r = 0$), thereby mitigating the overfitting and underfitting issues. Specifically, to show the robustness of our $\mathcal{L}_{rpd}$, we plot the loss curves of InfoNCE, MAE, and $\mathcal{L}_{rpd}$ with different $r$ as shown in Fig.5(a). We can observe that our loss and gradient are between that of InfoNCE and MAE. It indicates that when $0 < r < 1$, our $\mathcal{L}_{rpd}$ reduces the losses of hard sample pairs (*i.e.*, the mismatched prototypes), which could pay more attention to them compared with InfoNCE. Compared to MAE, our RPDL

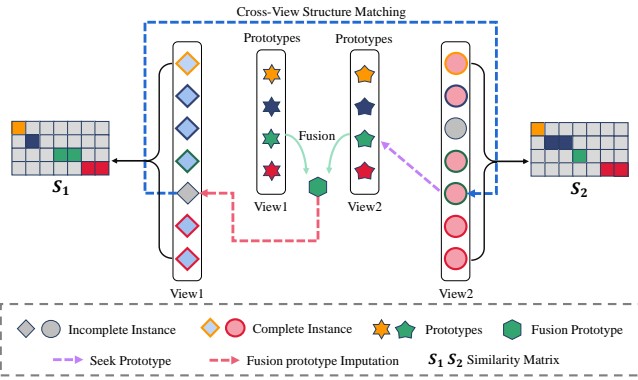

**Figure 4: The pipeline of prototype fusion-based imputation strategy.**

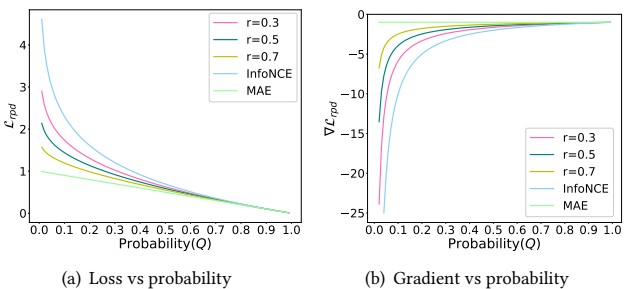

(a) Loss vs probability (b) Gradient vs probability

**Figure 5: The impact of different $r$ on the loss and gradient.**

can focus on information sample pairs discriminatively rather than treating them equally.

To further analyze the robustness of the proposed RPDL, we calculate the gradient $\nabla \mathcal{L}_{rpd}$ of our loss $\mathcal{L}_{rpd}$ as follows:

$$\frac{\partial \mathcal{L}_{rpd}}{\partial Q} = \frac{(1-Q)^r}{(-\log Q)^r}\left(\frac{r\log Q}{1-Q} - \frac{1-r}{Q}\right). \tag{11}$$

Then, we plot the gradient curve as shown in Fig.5(b). Clearly, compared with InfoNCE, when handling hard samples, the gradient magnitude (*i.e.*, $\nabla \mathcal{L}_{rpd}$) of our loss (*i.e.*, $0 < r < 1$) is smaller than InfoNCE. It indicates that the model prioritizes considering clean samples rather than noisy ones, thereby enhancing robustness against noise and mitigating overfitting issues. On the other hand, compared to MAE, the gradient magnitude of our loss is greater than MAE. It indicates that our loss can treat each sample discriminatively, thereby paying more attention to the rich information between the samples and effectively handle with underfitting. In general, by adjusting the value $r$, our loss could seek a balance to alleviate the overemphasis on hard samples and focus more on easy samples, thus enhancing robustness.

## 4 Experiments

### 4.1 Datasets

Four widely used multi-view datasets are chosen to validate the proposed RPCIC. Concretely, **Caltech101-7** [2] consists of 1,400 samples from seven categories. Each image is characterized by five types of features, *i.e.*, Pixel, Fourier, Profile, Zer, Kar, and Mor.

**HandWritten** [8] comprises 2,100 samples belonging to ten categories corresponding to digits from 0 to 9. Each sample is described by six types of features, *i.e.*, Pixel, Fourier, Profile, Zer, Kar, and Mor. **ALOI-100** [4] contains 10,800 object images belonged to 100 classes. We extract HSB, RGB, Colorsim, and Haralick features from these images to construct multi-view data, respectively. **YouTube-Face10** [34] is a facial video database with an extensive collection of 38,654 samples from 10 subjects. We use different feature extractors to obtain LBP, HOG, GIST, and Gabor features for each image, respectively.

### 4.2 Comparison Methods

To evaluate the effectiveness of our method, we compare RPCIC with 11 state-of-the-art clustering methods, including UEAF [33], IMSC_AGL [29], GIMC_FLSD [31], COMPLETER [14], SURE [40], DSIMVC [23], DIMVC [38], ProImp [9], DCP [13], APADC [37], and CPSPAN [7]. To comprehensively evaluate these comparison methods, we employ three widely used evaluation metrics, including accuracy (ACC), normalized mutual information (NMI), and f-measure (F-mea).

### 4.3 Experimental Settings

We perform our proposed RPCIC using PyTorch 1.8.1 and conduct all experiments on Linux with an NVIDIA 3090 GPU and 32GB RAM. For our method, we use the fully connected networks to achieve autoencoders for all views with the layer dimensions of $d_v$-1024-512-256 (256-512-1024–$d_v$) as the encoder (decoder), where $d_v$ represents the feature dimension of each view. In the experiment, we pre-train and fine-tune the model for 100 epochs with the learning rate of 0.0005 and the learning rate of 0.0001, respectively. In addition, we set the batch size as 256 and employ the Adam optimizer to optimize the model. ReLU is adopted for the activation function.

To show the performance of handling missing multi-view data, we set the missing rate to [0.1, 0.3, 0.5, 0.7, 0.9]. On the ALOI dataset, we set the parameters $\alpha$ and $\beta$ to 0.001 and 0.001, respectively. On the other datasets, we set them to 0.0001 and 0.01, respectively. The parameter $r$ is set to 0.5. Moreover, we set the prototype rate to 0.3 according to the experiments of prototype rate analysis.

### 4.4 Experimental comparative results

Table 1 shows the clustering results of different IMVC methods under three metrics on four datasets with different missing rates, where O/M represents out-of-memory. In addition, to further demonstrate the superiority of our RPCIC, we plot the ACC curves with different missing rates from 0 to 0.9 as shown in Fig. 6. From the table and figure, we have the following observations:

- Obviously, the proposed RPCIC achieves competitive performance compared with other IMVC methods in most cases. Especially, for high missing rates, RPCIC significantly outperforms these competitors for all metrics. For example, on the Caltech101-7 dataset with the missing rate of 0.9, RPCIC outperforms the second best by 6.43%, 7.99%, and 6.49%, respectively. This indicates that RPCIC can effectively leverage prototypes to imputation incomplete multi-view data, thereby capturing a better clustering structure.

**Table 1: Performance comparison across four datasets with five distinct missing rates. The best and second results are highlighted in red and blue, respectively.**

| | Missing_rates | 0.1 | | | 0.3 | | | 0.5 | | | 0.7 | | | 0.9 | | |
|---|---|---|---|---|---|---|---|---|---|---|---|---|---|---|---|---|
| | Metrics | ACC | NMI | F-mea | ACC | NMI | F-mea | ACC | NMI | F-mea | ACC | NMI | F-mea | ACC | NMI | F-mea |
| Caltech101-7 | UEAF(AAAI'19) | 42.43 | 29.49 | 33.42 | 38.50 | 25.94 | 30.33 | 36.79 | 22.79 | 27.64 | 37.43 | 24.31 | 29.45 | 34.64 | 23.57 | 27.54 |
| | IMSC_AGL(TCYB'20) | 75.36 | 70.08 | 68.08 | 64.56 | 66.38 | 64.03 | 73.50 | 68.57 | 66.50 | 61.13 | 59.83 | 58.35 | 59.43 | 61.45 | 63.15 |
| | GIMC_FLSD(TCYB'20) | 55.06 | 39.91 | 43.42 | 56.43 | 40.71 | 43.25 | 52.80 | 38.40 | 41.61 | 48.47 | 37.02 | 39.30 | 49.96 | 39.02 | 40.75 |
| | COMPLETER(CVPR'21) | 54.24 | 59.28 | 53.72 | 68.94 | 60.37 | 66.80 | 58.13 | 63.53 | 56.12 | 55.53 | 56.79 | 53.48 | 23.04 | 18.21 | 18.76 |
| | SURE(TPAMI'22) | 82.00 | 76.89 | 81.86 | 73.79 | 67.83 | 73.16 | 75.86 | 69.66 | 74.45 | 75.64 | 65.67 | 75.11 | 48.29 | 38.11 | 48.07 |
| | DSIMVC(ICLR'22) | 67.44 | 57.97 | 67.55 | 68.23 | 58.14 | 68.19 | 52.36 | 46.18 | 52.15 | 43.91 | 38.63 | 42.53 | 32.11 | 19.18 | 28.20 |
| | DIMVC(AAAI'22) | 82.21 | 67.22 | 66.31 | 77.32 | 71.50 | 62.26 | 75.21 | 56.09 | 55.62 | 60.43 | 41.47 | 46.34 | 60.64 | 56.20 | 53.18 |
| | ProImp(IJCAI'23) | 75.17 | 65.40 | 75.02 | 81.69 | 69.34 | 81.58 | 74.94 | 63.85 | 74.90 | 71.97 | 58.61 | 72.10 | 67.31 | 55.49 | 66.98 |
| | DCP(PAMI'23) | 56.53 | 58.70 | 55.93 | 54.66 | 54.92 | 53.47 | 55.37 | 59.64 | 52.51 | 33.66 | 38.16 | 28.95 | 35.83 | 23.27 | 8.18 |
| | APADC(TIP'23) | 73.70 | 68.54 | 73.92 | 75.57 | 71.66 | 74.35 | 83.46 | 75.07 | 82.18 | 59.14 | 56.00 | 53.44 | 43.66 | 40.40 | 37.38 |
| | CPSPAN(CVPR'23) | 89.36 | 82.00 | 89.05 | 86.00 | 78.04 | 85.68 | 86.00 | 75.40 | 85.44 | 85.57 | 75.74 | 85.22 | 79.43 | 68.99 | 79.06 |
| | **RPCIC(Ours)** | **90.29** | **82.79** | **89.94** | **90.71** | **83.11** | **90.43** | **90.29** | **83.35** | **90.03** | **88.93** | **81.33** | **88.46** | **85.86** | **76.98** | **85.55** |
| HandWritten | UEAF(AAAI'19) | 44.10 | 54.35 | 40.25 | 60.90 | 56.33 | 49.39 | 51.60 | 49.34 | 41.97 | 44.00 | 37.07 | 34.29 | 31.65 | 26.35 | 25.50 |
| | IMSC_AGL(TCYB'20) | 84.56 | 79.95 | 79.87 | 77.97 | 77.52 | 73.13 | 79.11 | 72.12 | 70.38 | 78.60 | 70.35 | 68.54 | 78.78 | 70.07 | 69.53 |
| | GIMC_FLSD(TCYB'20) | 22.16 | 24.50 | 25.94 | 24.60 | 22.94 | 24.43 | 18.99 | 10.83 | 18.69 | 22.01 | 14.33 | 19.07 | 23.89 | 12.67 | 18.59 |
| | COMPLETER(CVPR'21) | 76.11 | 79.53 | 71.66 | 73.76 | 76.16 | 73.32 | 72.50 | 71.57 | 73.16 | 81.39 | 78.50 | 80.83 | 20.97 | 24.56 | 19.93 |
| | SURE(TPAMI'22) | 66.85 | 58.02 | 66.28 | 73.70 | 63.31 | 73.26 | 73.35 | 63.22 | 72.98 | 69.85 | 59.95 | 68.36 | 51.80 | 44.00 | 50.62 |
| | DSIMVC(ICLR'22) | 79.00 | 74.81 | 79.08 | 78.27 | 74.15 | 78.33 | 71.17 | 68.53 | 71.09 | 69.06 | 64.57 | 68.95 | 45.46 | 42.85 | 44.38 |
| | DIMVC(AAAI'22) | 63.15 | 58.72 | 61.85 | 60.15 | 53.35 | 55.88 | 59.75 | 48.78 | 57.01 | 41.45 | 29.83 | 38.55 | 30.65 | 22.07 | 23.10 |
| | ProImp(IJCAI'23) | 84.64 | 82.07 | 84.37 | 83.12 | 78.78 | 82.83 | 80.25 | 74.62 | 68.53 | 80.78 | 70.96 | 80.44 | 77.35 | 66.85 | 76.93 |
| | DCP(PAMI'23) | 72.95 | 75.33 | 54.78 | 72.00 | 72.11 | 51.07 | 71.46 | 74.16 | 54.69 | 58.43 | 63.44 | 32.10 | 33.14 | 31.08 | 8.25 |
| | APADC(TIP'23) | 81.67 | 83.80 | 80.02 | 81.65 | 83.75 | 79.56 | 78.37 | 79.86 | 75.26 | 57.03 | 65.26 | 51.90 | 55.61 | 55.05 | 49.58 |
| | CPSPAN(CVPR'23) | 88.75 | 81.94 | 88.68 | 91.05 | 83.29 | 91.08 | 78.85 | 78.85 | 77.05 | 87.55 | 78.15 | 87.51 | 80.30 | 77.03 | 78.84 |
| | **RPCIC(Ours)** | **91.95** | **84.85** | **91.94** | **91.25** | **83.86** | **91.24** | **90.25** | **82.35** | **90.29** | **89.15** | **81.02** | **89.14** | **88.20** | **79.78** | **88.16** |
| ALOI-100 | UEAF(AAAI'19) | 39.86 | 64.06 | 26.39 | 34.18 | 59.48 | 23.19 | 32.48 | 53.93 | 17.55 | 27.03 | 45.44 | 9.69 | 22.31 | 37.25 | 6.25 |
| | IMSC_AGL(TCYB'20) | O/M | O/M | O/M | O/M | O/M | O/M | O/M | O/M | O/M | O/M | O/M | O/M | O/M | O/M | O/M |
| | GIMC_FLSD(TCYB'20) | 39.53 | 59.89 | 25.88 | 32.06 | 52.79 | 18.10 | 25.26 | 43.28 | 10.24 | 19.76 | 37.06 | 6.49 | 15.72 | 31.42 | 4.25 |
| | COMPLETER(CVPR'21) | 56.64 | 79.03 | 54.28 | 45.59 | 73.31 | 42.92 | 33.72 | 65.82 | 31.49 | 38.74 | 68.12 | 35.87 | 29.20 | 58.31 | 27.34 |
| | SURE(TPAMI'22) | 33.71 | 71.34 | 30.09 | 29.82 | 67.75 | 25.07 | 28.94 | 67.29 | 23.25 | 36.19 | 64.92 | 22.06 | 20.89 | 57.93 | 18.20 |
| | DSIMVC(ICLR'22) | 64.72 | 72.90 | 45.64 | 63.75 | 70.47 | 44.57 | 60.26 | 71.93 | 42.48 | 57.02 | 66.60 | 39.68 | 55.48 | 63.13 | 45.57 |
| | DIMVC(AAAI'22) | 67.50 | 81.29 | 63.56 | 59.70 | 73.81 | 57.87 | 55.17 | 68.91 | 52.59 | 54.32 | 67.78 | 51.71 | 34.36 | 49.66 | 35.72 |
| | ProImp(IJCAI'23) | 67.17 | 81.71 | 66.31 | 43.22 | 70.21 | 42.78 | 32.44 | 64.33 | 33.06 | 28.64 | 61.13 | 29.83 | 30.39 | 57.60 | 31.04 |
| | DCP(PAMI'23) | 50.77 | 78.35 | 43.11 | 46.67 | 73.43 | 44.48 | 41.80 | 70.91 | 39.82 | 39.24 | 68.32 | 36.84 | 29.20 | 66.01 | 37.27 |
| | APADC(TIP'23) | 35.81 | 59.96 | 33.22 | 34.54 | 58.11 | 30.58 | 29.02 | 54.74 | 27.11 | 23.39 | 50.79 | 23.62 | 20.33 | 48.39 | 21.09 |
| | CPSPAN(CVPR'23) | 71.69 | 85.30 | 68.55 | 67.96 | 82.19 | 66.34 | 66.62 | 82.39 | 63.74 | 65.44 | 82.13 | 62.50 | 58.80 | 74.95 | 56.34 |
| | **RPCIC(Ours)** | **74.50** | **88.61** | **71.85** | **71.85** | **86.97** | **69.25** | **69.68** | **85.85** | **68.32** | **69.23** | **86.28** | **66.63** | **66.62** | **84.14** | **64.11** |
| YouTubeFace10 | UEAF(AAAI'19) | O/M | O/M | O/M | O/M | O/M | O/M | O/M | O/M | O/M | O/M | O/M | O/M | O/M | O/M | O/M |
| | IMSC_AGL(TCYB'20) | O/M | O/M | O/M | O/M | O/M | O/M | O/M | O/M | O/M | O/M | O/M | O/M | O/M | O/M | O/M |
| | GIMC_FLSD(TCYB'20) | O/M | O/M | O/M | O/M | O/M | O/M | O/M | O/M | O/M | O/M | O/M | O/M | O/M | O/M | O/M |
| | COMPLETER(CVPR'21) | 61.77 | 67.59 | 62.14 | 55.74 | 63.20 | 53.74 | 55.80 | 64.05 | 56.78 | 54.90 | 61.08 | 54.51 | 48.68 | 57.04 | 48.58 |
| | SURE(TPAMI'22) | 64.59 | 71.91 | 61.95 | 66.04 | 70.67 | 63.72 | 72.14 | 77.07 | 69.54 | 67.10 | 72.23 | 65.90 | 57.23 | 70.10 | 56.01 |
| | DSIMVC(ICLR'22) | 71.08 | 74.66 | 73.46 | 71.94 | 75.35 | 74.45 | 71.03 | 74.81 | 73.74 | 69.35 | 74.21 | 71.71 | 64.89 | 70.81 | 67.54 |
| | DIMVC(AAAI'22) | 65.54 | 68.26 | 64.93 | 68.05 | 56.35 | 68.18 | 63.94 | 50.18 | 60.06 | 63.21 | 47.16 | 62.78 | 60.37 | 41.87 | 59.80 |
| | ProImp(IJCAI'23) | 53.47 | 59.11 | 52.21 | 55.71 | 63.90 | 54.92 | 53.82 | 62.24 | 52.02 | 51.48 | 58.21 | 49.58 | 47.16 | 57.32 | 45.90 |
| | DCP(PAMI'23) | 62.04 | 69.81 | 63.47 | 54.28 | 63.35 | 52.33 | 56.50 | 62.43 | 58.22 | 58.68 | 65.42 | 59.71 | 52.18 | 59.63 | 51.62 |
| | APADC(TIP'23) | 65.66 | 68.85 | 67.18 | 72.50 | 74.82 | 74.63 | 69.43 | 70.75 | 68.45 | 63.69 | 64.40 | 60.05 | 56.02 | 57.87 | 48.16 |
| | CPSPAN(CVPR'23) | 73.94 | 79.21 | 71.27 | 68.12 | 79.34 | 64.17 | 72.68 | 80.05 | 72.93 | 69.58 | 75.13 | 67.59 | 64.92 | 77.76 | 62.27 |
| | **RPCIC(Ours)** | **74.45** | **79.14** | **73.05** | **73.58** | **75.37** | **74.86** | **74.26** | **74.76** | **73.64** | **70.77** | **76.04** | **71.95** | **69.93** | **76.24** | **68.66** |

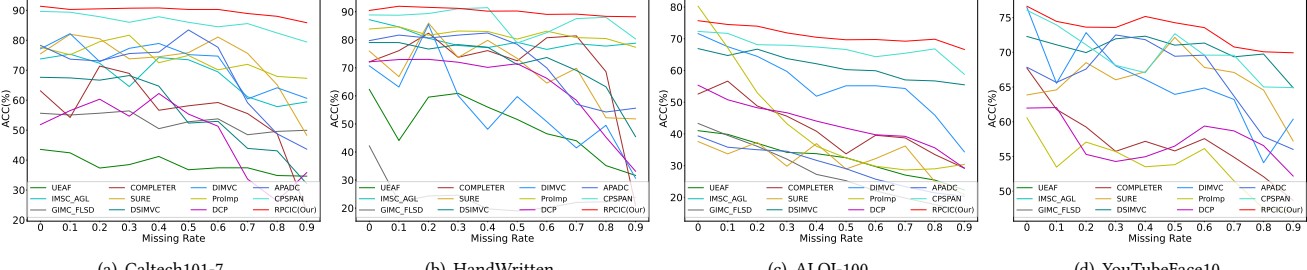

(a) Caltech101-7    (b) HandWritten    (c) ALOI-100    (d) YouTubeFace10

**Figure 6: The ACC performance results on four datasets with different missing rates.**

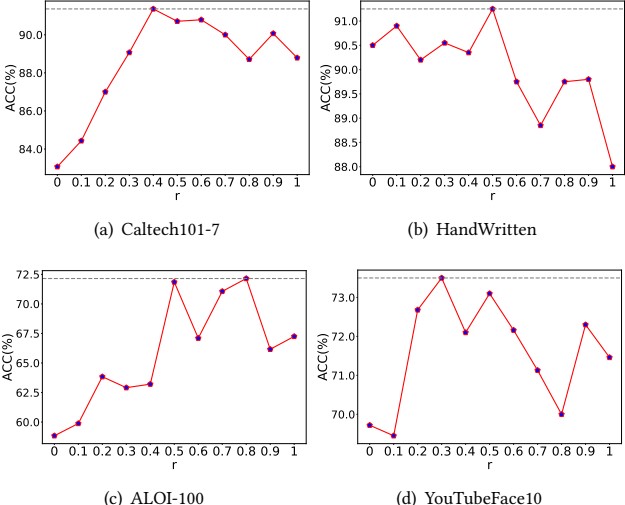

(a) Caltech101-7

(b) HandWritten

(c) ALOI-100

(d) YouTubeFace10

**Figure 7: Parameter analysis of *r* with 0.3 missing rate.**

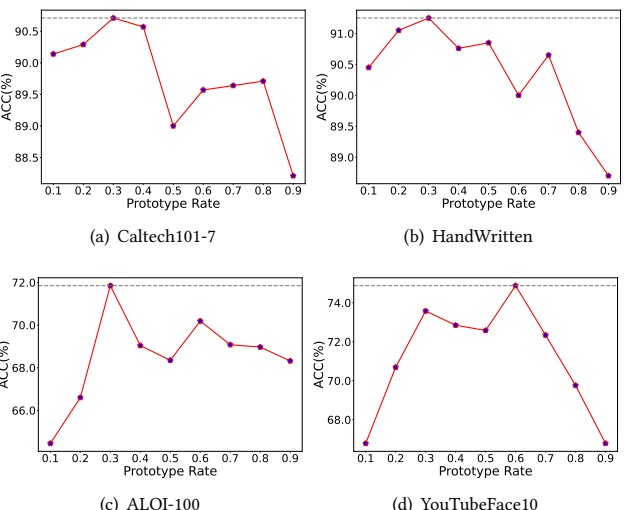

(a) Caltech101-7

(b) HandWritten

(c) ALOI-100

(d) YouTubeFace10

**Figure 8: The ACC performance with different prototype rates on four datasets with the missing rate 0.3.**

- As the missing rate increases, the clustering performance of almost all IMVC methods decreases. It is worth noting that under a missing rate of 0.9, the performance of our method is comparable to that of CPSPAN under a missing rate of 0.7. This is attributed to the ability of RPCIC to robustly learn high-quality prototypes with limited samples, and accurately recover missing data by our prototype imputation strategy.
- As the missing rate increases, most IMVC methods show large performance fluctuations, while our RPCIC remains relatively stable. This trend indicates that robust prototype completion can compensate for the missing information from different views.
- We observe that on YouTubeFace10, some scenarios with high missing rates outperform those with lower missing rates. This could be because multi-view data has varying

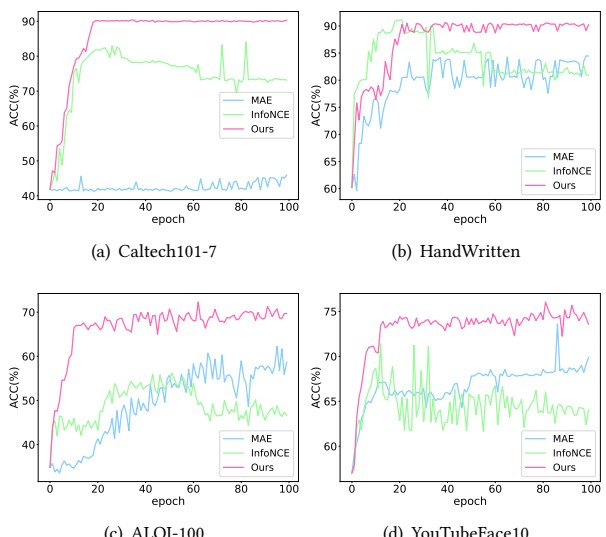

(a) Caltech101-7

(b) HandWritten

(c) ALOI-100

(d) YouTubeFace10

**Figure 9: Robust analysis on the four datasets with the missing rate 0.5.**

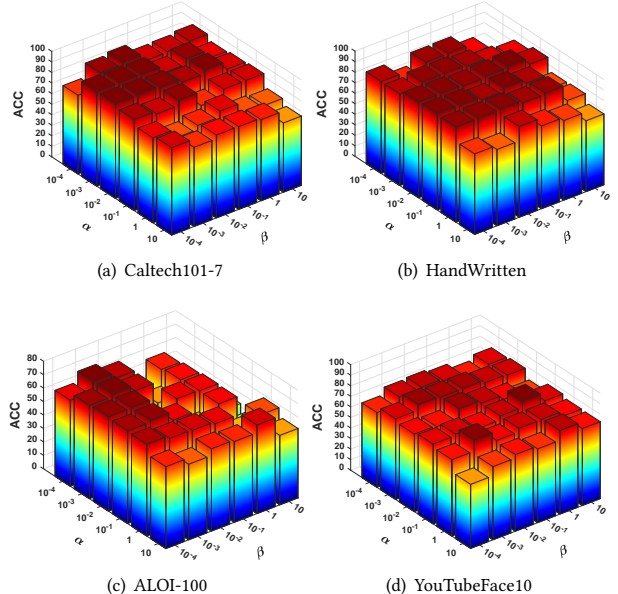

(a) Caltech101-7

(b) HandWritten

(c) ALOI-100

(d) YouTubeFace10

**Figure 10: Parameter sensitivity analysis on four datasets with the missing rate 0.3.**

quality across different views. Under high missing rates, a significant portion of low-quality data is discarded, leaving behind data with better feature information. Consequently, the extracted features and prototypes exhibit higher quality.

### 4.5 Robustness Experiments

To evaluate the robustness of the proposed RPCIC, we compare it with two widely-used losses, *i.e.*, InfoNCE and MAE. Specifically, we set the missing rate to 0.3 and draw the ACC performance curves as shown in Fig.9. From the figure, we can observe that the typical

**Table 2: Ablation studies on four datasets with different missing rates, where '√' indicates the used component.**

| Datasets | | | Caltech101-7 | | | | HandWritten | | | | ALOI-100 | | | | YouTubeFace10 | | | |
|---|---|---|---|---|---|---|---|---|---|---|---|---|---|---|---|---|---|---|
| $\mathcal{L}_{rec}$ | $\mathcal{L}_{ccl}$ | $\mathcal{L}_{rpd}$ | 0.1 | 0.3 | 0.5 | 0.7 | 0.1 | 0.3 | 0.3 | 0.7 | 0.1 | 0.3 | 0.5 | 0.7 | 0.1 | 0.3 | 0.5 | 0.7 |
| √ | | | 74.36 | 73.71 | 65.36 | 56.93 | 78.40 | 78.35 | 75.25 | 73.35 | 66.35 | 63.17 | 62.19 | 61.02 | 68.23 | 68.46 | 65.34 | 59.68 |
| √ | √ | | 88.50 | 88.79 | 82.43 | 79.57 | 90.20 | 89.30 | 87.55 | 77.75 | 72.41 | 68.45 | 67.3 | 63.14 | 70.39 | 71.06 | 68.65 | 64.32 |
| √ | | √ | 89.79 | 88.00 | 84.86 | 83.14 | 89.20 | 86.45 | 84.20 | 80.55 | 72.48 | 70.65 | 66.24 | 67.36 | 70.89 | 69.35 | 70.36 | 68.20 |
| √ | √ | √ | **90.29** | **90.71** | **90.29** | **88.93** | **91.95** | **91.25** | **90.25** | **89.15** | **74.50** | **71.85** | **69.68** | **69.32** | **74.45** | **73.58** | **74.26** | **70.77** |

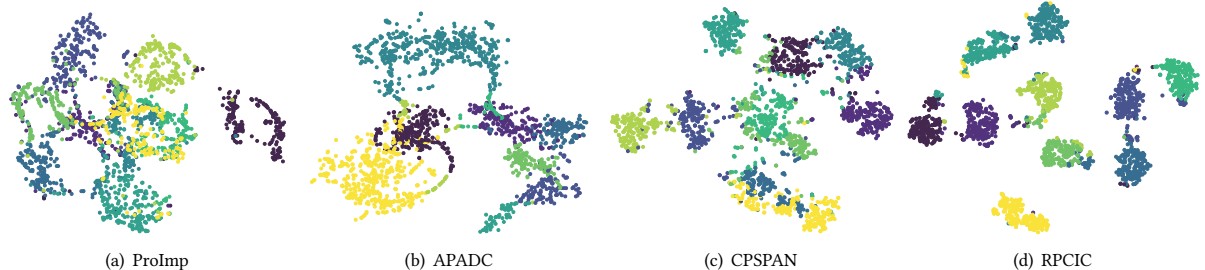

(a) ProImp         (b) APADC         (c) CPSPAN         (d) RPCIC

**Figure 11: Visualization results on HandWritten with missing rates 0.9.**

InfoNCE suffers from the overfitting problem. In other words, the performance first increases and then decreases due to overemphasis on hard samples (*i.e.*, false positives). Although robust MAE loss can alleviate overfitting to achieve more stable performance, it suffers from the underfitting problem, thereby leading to lower clustering performance. For our RPCIC, it achieves a steady improvement in the ACC performance, which shows strong robustness to overcome the overfitting problem caused by prototype noisy correspondence. To further analyze the impact of $r$, we plot the ACC performance curves with different $r$ from 0 to 1 on four datasets with the missing rate 0.3. As shown in Fig.7, the ACC scores first increase and then decreases as increasing $r$. The results show that either too high or too low $r$ value is not conducive to clustering. With a small $r$, the RPD loss tends to the InfoNCE loss, thereby overemphasizing hard samples. While a large $r$ tends to the MAE loss, thereby leading to underfitting. Experimentally, we suggest setting $r$ to 0.5.

### 4.6 Influence of Different Prototype Rates

To explore the impact of different numbers of prototypes on clustering performance, we conduct experiments under different prototype rates from 0.1 to 0.9. As shown in Fig.8, the appropriate number of prototypes facilitates clustering. Specifically, insufficient prototypes could result in various class structures being mapped to a single prototype, causing an overemphasis on commonalities among data while neglecting the distinctions between different cluster structures. On the contrary, excessive prototypes could make each prototype contain little cluster information, failing to adequately capture the commonalities of cluster structures. According to our experiments, we choose the prototype rate 0.3 on four datasets.

### 4.7 Ablation Study

To study the importance of each component, we conduct an ablation analysis of three versions of our RPCIC on four datasets with different missing rates (*i.e.*, 0.1, 0.3, 0.5, and 0.7). As shown in Table 2, the loss of CCL or RPD is insufficient to achieve the best performance, and the effectiveness is instability under the high missing rate. When all three losses are utilized, we can obtain the

best performance. It demonstrates that the loss CCL could improve cross-view consistency and discriminability, and the loss RPD could effectively alleviate prototype noisy correspondence.

### 4.8 Parameter Sensitivity Analysis

Our objective loss mainly contains two trade-off parameters, $\alpha$ and $\beta$. To verify their effectiveness, we perform parameter analysis by setting the two parameters to vary from $10^{-4}$ to $10^{1}$. As shown in Fig.10, we can observe that too large or too small parameter values are not beneficial to clustering. According to the parameter experiments, we can obtain the optimal parameter values.

### 4.9 Visualization

We visualize the clustering effect of our method and three state-of-the-art methods (*i.e.*, ProImp, APADC, CPSPAN ) on HandWritten with the missing rate 0.9 as shown in Fig.11. For three comparison methods, the boundaries between different classes become blurred, and the distribution of instances within the same class becomes more dispersed. However, our RPCIC shows that intra-class distributions remain compact and inter-class boundaries are clear. It could be attributed to the losses of CCL and RPD, which could reduce the heterogeneity gap and overcome the prototype noisy correspondence problem.

### 5 Conclusion

In this paper, we present an RPCIC framework to address the challenging issue in IMVC tasks, *i.e.*, prototype noisy correspondence. To tackle this issue, we employ a cross-view contrastive learning strategy to maximize multi-view consistency, thereby narrowing the multi-view heterogeneous gap. Subsequently, to deal with the noisy correspondence problem in generated prototypes, we propose a robust prototype discriminative learning loss to robustly learn from prototype noisy correspondence, thereby mitigating the overfitting and underfitting problems. Finally, for missing data imputation, we propose a prototype fusion-based imputation strategy. Extensive experiments demonstrate the superiority and robustness of our RPCIC compared with these competitors.

# Acknowledgments

This work was supported by the Base Strengthening Program of National Defense Science and Technology (Grant No. 2022-JCJQ-JJ-0292).

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
