# OpenReview forum: "Robust Prototype Completion for Incomplete Multi-view Clustering"
_acmmm.org/ACMMM/2024/Conference — MM2024 Poster_

### Official Review · Reviewer_FTSB · 2024-04-30

**Rating:** 6
**Confidence:** 3

**Summary:**

In this paper, authors propose an incomplete multi-view clustering algorithm that considers prototype noisy correspondence, i.e., Robust Prototype Completion for Incomplete Multi-view Clustering (RPCIC). The authors propose robust prototype discriminative learning to alleviate the prototype noisy correspondence. Moreover, they propose a prototype fusion-based imputation strategy to recover missing data. Multiple benchmark experiments verify its effectiveness and robustness.

**Strengths:**

1: This is the first paper to consider the issue of prototype noisy correspondence in the IMVC task, it enhances the practical significance of the method.
2: The cross-view contrastive learning strategy reduces the heterogeneity among multi-view data and enhances the consistency of the multi-view data.
3: This paper proposes the Robust Prototype Discriminative Learning loss, enhancing the model's robustness of prototype noisy correspondence.
4: This paper proposes a prototype fusion-based imputation strategy, which can effectively recover missing data with high quality.
5: This paper conducts abundant experiments to verify the effectiveness of RPCIC, and the experiments are sufficient and comprehensive.

**Limitations:**

1: The description of the prototype fusion strategy is overly simplistic. What is the prototype fusion strategy's purpose?
2: This paper lacks an analysis of the computational complexity of the algorithm.
3: To my knowledge, noisy correspondence learning has been studied in some other tasks. However, the authors do not give some discussions.

**Suitability:**

3

---

### Official Review · Reviewer_wJgj · 2024-05-25

**Rating:** 5
**Confidence:** 4

**Summary:**

This paper proposes a method for incomplete multi-view clustering. Specifically, it proposes cross-view contrastive learning strategy to learn the consistency of cross-view representations and robust prototype discriminative learning strategy to address prototype noisy correspondence. Based upon these, the missing data are recovered by a prototype fusion-based strategy.

**Strengths:**

1. The overall structure of this paper is clear and logical. Collectively, the overall quality of this paper is high and readable.
2. The proposed method can still achieve perfect recovery performance in the case of high missing rate, providing evidence in dealing with missing data effectively.
3. The experimental design is rigorous, and the authors use a large number of comparative experiments and statistical analyses to ensure the high reliability of the results. These experimental findings provide robust support for the main conclusions of the paper, enhancing its scientific rigor and credibility.

**Limitations:**

1. The performance is sensitive to the parameter settings, especially the parameter $r$.
2. As far as I know, prototype learning can improve computational efficiency, but the authors do not mention this in the paper.

**Suitability:**

3

---

### Official Review · Reviewer_JcwX · 2024-05-25

**Rating:** 5
**Confidence:** 3

**Summary:**

This paper introduces the Robust Prototype Completion technique for Incomplete Multi-view Clustering. It harnesses the pair-observed representations from comprehensive data sets for training purposes. Additionally, it addresses the heterogeneous characteristics of multi-view data through a cross-view contrastive learning approach, while effectively tackling the issue of noisy prototype correspondence via a robust prototype discriminative learning strategy. Both theoretical analysis and experimental validation demonstrate the efficacy and usefulness of the proposed method.

**Strengths:**

1. Writing: The paper is easy to read, and generally well written.
2. Novelty: The cross-view contrastive learning strategy effectively eradicates the heterogeneity present in multi-view data.
3. Experimental: This paper utilizes a theoretical analysis and practically demonstrates the robustness of the robust prototype discriminative learning strategy.

**Limitations:**

1. On page 3, this method consists of two phases: warm-up and fine-tune. However, it is not explicitly explained whether the within-view reconstruction loss is used in the second stage.
2. The proposed method uses trained data to recover missing views, but doesn't use the restored missing views in continue training. It is not clearly explained in the manuscripts why training with recovered missing views is not considered.
3. In Section EXPERIMENTS, authors are advised to add a comparison of computational complexity.
4. In Section REFERENCES, some references lack information such as page numbers. The authors are advised to read through the paper carefully to make sure there are no errors in detail.

**Suitability:**

3

---

### Meta-Review · Area_Chair_AQV8 · 2024-07-06

**Recommendation:** Accept (Poster)
**Confidence:** 4

**Metareview:**

This paper proposes Robust Prototype Completion for Incomplete Multi-view Clustering (RPCIC), which reduces the impact of noisy correspondence in prototypes. Initially, RPCIC employs a cross-view contrastive learning module to obtain consistent feature representations across different views. Subsequently, a robust contrastive loss is devised for the produced prototypes to alleviate the influence of noisy correspondence. Finally, a prototype fusion-based strategy is used to complete the missing data.

Reviewers feedbacks are positive: WA/WA/A; the writing is clear, the contribution is sufficient novel and experiments are suitable.